# Digital Leadership’s Influence on Individual Creativity and Employee Performance: A View through the Generational Lens

**DOI:** 10.3390/bs14010003

**Published:** 2023-12-20

**Authors:** Volkan Öngel, Ayşe Günsel, Gülşah Gençer Çelik, Erkut Altındağ, Hasan Sadık Tatlı

**Affiliations:** 1Department of International Trade and Finance, Faculty of Economics and Administrative Sciences, Istanbul Beykent University, Istanbul 34075, Türkiye; volkanongel@beykent.edu.tr; 2Department of Management, Faculty of Business Administration, Kocaeli University, Kocaeli 41380, Türkiye; 3Department of Business Management, Vocational Higher School, Istanbul Beykent University, Istanbul 34075, Türkiye; gulsahg@beykent.edu.tr; 4Department of Business Management, Faculty of Economics and Administrative Sciences, Istanbul Beykent University, Istanbul 34075, Türkiye; erkutaltindag@beykent.edu.tr (E.A.); hasantatli@beykent.edu.tr (H.S.T.)

**Keywords:** digital leadership, individual creativity, job performance, generations

## Abstract

Today’s digitalized business atmosphere places significant emphasis on digital leadership, as digital transformation can only be successful for leaders who are capable of overseeing the entire digitalization process. In this study, we examine the employee-level outputs of digital leadership in terms of individual creativity and employee performance. Based on the data of 348 employees and by using PLS-SEM, we find that (i) digital leadership is a significant antecedent of individual creativity, and (ii) individual creativity fully mediates the relationship between digital leadership and employee performance. When digital leadership abilities are a matter of concern, it becomes necessary to mention the importance of generational differences between those leaders, as OB studies also underline the generational differences toward new technologies. Hence, we also conducted a multigroup analysis (MGA) to determine if those interrelationships among digital leadership, individual creativity, and employee performance differ due to the generations of the managers. Surprisingly, the MGA results reveal minor differences. The findings of this study highlight the importance of embracing digital leadership in fostering a creative and high-performing work environment and provide insights for organizations aiming to harness the unique strengths and talents of their multigenerational workforce. Digital leaders who foster a culture of innovation, adaptability, and open communication can inspire employees to think creatively and come up with novel ideas. By leveraging digital tools and providing a supportive environment, digital leaders can also enhance employee creativity and performance.

## 1. Introduction

As we navigate the aftermath of the pandemic, the world is witnessing a new reality shaped by the pervasive influence of digital technologies in all aspects of human life [1,2,3,4]. Recent technological advancements are revolutionizing existing processes and business models [5,6]. Consequently, this post-COVID-19 era requires updated skills to confront digital transformation [7,8]. With global influences, both macro and micro, making business situations more intricate, traditional leadership approaches and skills need to be enhanced in the Revolution 4.0 era [9]. Despite the fact that most companies today are developing new digital business strategies, only a few recognize the critical role of digital leadership in this process [10,11]. However, digital transformation can only be successful with leaders who are capable of overseeing the entire digitalization process [12]. Therefore, recent theory, research, and practice in management and organizational behavior place significant emphasis on digital leadership because firms need to “jump onto the digital transformation bandwagon” [13].

Digital leadership is the exploitation of digital assets to achieve organizational goals [11,14,15]. It involves combining a leader’s abilities with digital technologies [16]. Employees, by themselves, may realize the importance of the digital world; leaders, however, should deeply understand what it means and be more aware of Industry 4.0 and the forthcoming Revolution 5.0 [17]. Digital leaders should have the necessary set of skills, i.e., so-called digital leadership abilities, to bring value to their organizations in these hyperdynamic and digitalizing business environments [12,18,19]. They should also be able to (i) communicate digitally (i.e., digital communication skills) by Internet-based media, as digital leaders generally run their business through a digital artifact, typically a platform [20], and (ii) integrate culture and digital competence to utilize digital technologies [21,22].

Moreover, those leaders are responsible for selling the digital vision internally and externally and moving the organization to a more digitalized axis (i.e., vision toward digital technologies) [13]. As probably the most studied concept of OB, leadership has long been considered a key determinant of employee creativity and performance, e.g., references [23,24,25,26,27,28,29,30]. However, the studies examining the reflections of digital leadership on individual and organizational outcomes are scant [13], as digital leadership emerges as a relatively novel concept. To fill this gap, we have chosen to investigate the influence of digital leadership skills on employee outcomes, specifically focusing on creativity and employee performance. In this endeavor, we adopt a comprehensive 2D framework for assessing employee performance, encompassing tasks, and contextual dimensions, as elucidated in Sackett’s seminal work [31]. By exploring the antecedent role played by digital leadership skills in employee creativity and performance, our objective is to contribute to advancing the literature on digital leadership.

When digital leadership is a matter of concern, it becomes necessary to mention the importance of generational differences among those leaders, as OB studies also underline the generational differences toward new technologies. Today’s business environments entail a multigenerational workforce working side by side in general, particularly a multigenerational managerial force [32]. There are significant differences among those generations in terms of perceptions, stereotypes, and personalities [33,34]. Each generational cohort, e.g., Baby Boomers, Gen Xs, Gen Ys, and Gen Zs, has different attitudes toward new technologies [35]. For instance, Gen Y is commonly considered the “Net” generation, growing up with greater access to technology and being more prone to taking “digital” breaks from work, while older generations may not develop such positive attitudes toward new technologies [36]. Specifically, for Baby Boomers, digital technologies may be a source of techno-stress [37].

Since digital leadership involves integrating a leader’s abilities with digital technologies, we expect the generational cohorts to which leaders belong to influence how these leaders perform those digital abilities and how those abilities are reflected in employee outcomes [13]. Hence, this paper also seeks to enrich the nascent literature on generational differences in the ever-digitalizing workplace, aiming for a clearer understanding of how generational cohorts of managers, including Baby Boomers, Gen Xs, and Gen Ys, influence the interrelationship among digital leadership abilities, individual creativity, and employee performance.

This study is primarily driven by two crucial research questions: (i) the influence of digital leadership capabilities on individual creativity and employee performance and (ii) the variations in the impact of digital leadership capabilities on individual creativity and employee performance across different generations of managers. To date, we have not encountered any research that explores the interconnections among digital leadership, individual creativity, and employee performance from a generational standpoint. As far as we are aware, no comprehensive methodological framework has been developed for this approach. Theoretically, this study enriches OB theory by uncovering how digital leadership’s impact on employee outcomes varies based on the generational differences of managers. Practically, the proposed model (Figure 1) improves managers’ comprehension of boosting employee creativity and performance by leveraging digital leadership skills. Further, our model highlights potential disparities and shortcomings of older managers compared with their younger counterparts in adapting to the demands of the digital era, which may result in subpar employee outcomes, thus impeding the generation and implementation of creative ideas and superior performance.

## 2. Theoretical Background and Hypotheses Development

### 2.1. Digital Leadership

The COVID-19 pandemic, coupled with mobility restrictions and ongoing digital transformation, has accelerated the adoption of remote work, leading to fundamental changes in established management and social systems [38]. To effectively navigate the challenges of this new era and drive organizational progress, leaders must adopt a digital mindset and possess specific skills and abilities collectively known as “digital leadership” [39]. Digital leadership is a relatively new concept that has emerged with the advent of digital technologies and the subsequent transformation of businesses. It encompasses the skills and abilities of a leader to influence, motivate, and enable individuals to contribute effectively to an organization’s success through the appropriate utilization of digital resources. However, there remains a lack of clarity and consensus regarding the definition and content of digital leadership [40,41,42,43,44]. Some authors differentiate between digital leadership and leadership in the digital age or the digitalization process; others argue that the two concepts are interchangeable [42,45,46,47,48]. 

Moreover, the existing literature highlights the association of digital leadership with interactionist, transformational, empowering, and authentic leadership styles, suggesting their significance in achieving successful digital transformations [49,50,51]. Further, Sow and Aborbie [52] define digital leadership as a multileadership approach that consistently manages digital transformation processes and provides a competitive advantage, while Sheninger [53] defines a digital leader as someone who establishes direction, influences others, initiates sustainable change by accessing information, and establishes relationships to anticipate changes that will lead to future success. Larjovuori et al. [47] highlight the ability-based perspective of digital leadership.

Digital leaders must possess a specific set of skills called “digital leadership abilities,” which are crucial for adding value to organizations in today’s rapidly changing and digitalizing business environments [12,18,19]. These leaders must excel in various areas to navigate the digital landscape effectively. First, they must possess strong digital communication skills, thus enabling effective communication through Internet-based media, as digital leaders primarily operate using digital platforms [20]. Further, it is essential for digital leaders to possess effective communication and networking skills, not only within the organization but also with external parties, including stakeholders. In this vein, Harjoto and Wang [54] underscore that managers with strong networks are more likely to achieve superior outcomes.

Second, digital leaders should integrate organizational culture with digital competence to leverage digital technology effectively [21,22,55]. This integration process entails combining a deep understanding of an organization’s values, norms, and practices with the ability to harness the potential of digital tools and technologies.

Digital leaders also have a crucial responsibility to promote and sell the digital vision internally and externally, guiding their organizations toward a more digitalized future [13]. They must articulate a clear vision for utilizing digital technologies, align the entire organization around this vision, and drive the necessary changes to adapt and thrive in the digital era.

In summary, digital leaders require diverse skills, including effective digital communication, the integration of culture and digital competence, and the ability to shape and promote a digital vision. These capabilities empower leaders to effectively lead their organizations in dynamic and digital-centric business environments [12,13,18,19,20,21,22]. As digital transformation continues to reshape the business landscape, developing and honing these digital leadership abilities are essential for organizational- and individual-level outputs, such as employee creativity and performance, to thrive in the digital age.

### 2.2. Effects of Digital Leadership on Individual Creativity 

The extant literature emphasizes the crucial role of leadership as a primary driver of employee creativity [22]. In the digital age, digital leadership has emerged as a contemporary leadership style with direct implications for employee outcomes. First, digital leaders with superior digital communication skills [12,18,19] can effectively collaborate with employees through various digital channels. This collaboration facilitates open and transparent communication, allowing employees to share their ideas, thoughts, and creative insights easily. Indeed, utilizing online social networks, including electronic word of mouth (eWOM), has been demonstrated to possess significant potential in influencing and modifying individuals’ behaviors [56]. Therefore, by harnessing digital platforms, online social networks, and tools, digital leaders can promote brainstorming, idea sharing, and collaborative problem-solving, thus fostering a culture of innovation and creativity within the organization [20].

Second, digital leaders’ integration of organizational culture and digital competence [21,22] plays a vital role in nurturing employee creativity. When leaders prioritize digital competence and their organization’s cultural values, employees become equipped with the necessary knowledge, skills, and resources to explore and experiment with digital technologies. This integration enables employees to creatively leverage digital tools and platforms, thus unlocking new possibilities for innovative approaches toward work and problem-solving [21].

Furthermore, digital leaders who effectively promote and sell the digital vision [13] can inspire and motivate employees to embrace digitalization and think creatively about their roles and responsibilities. Leaders establish a sense of purpose and direction by articulating a clear vision and demonstrating how digital technologies can enhance individual and team performance. This empowers employees to explore new ideas, take calculated risks, and think beyond traditional boundaries, ultimately fostering a more creative work environment [13].

Hence, we expect that possessing digital leadership abilities contributes to employee creativity. Through effective digital communication, the integration of culture and digital competence, and the promotion of a digital vision, digital leaders create an environment that nurtures and encourages employee creativity.

**Hypothesis** **H1.**
*Digital leadership is positively and significantly related to individual creativity.*


### 2.3. Effects of Digital Leadership on Job Performance

Research suggests that digital leadership has emerged as a significant factor influencing employee behavior and performance in task and contextual domains. Notably, leaders leverage digital technologies to enhance communication, collaboration, and knowledge-sharing, ultimately leading to improved task performance [57]. For instance, Choi and Cho [58] demonstrated that leaders who utilize online networks have successfully led employees to achieve better performance outcomes by emphasizing safety management standards. Hence, by effectively utilizing digital platforms, social media, and other tools, digital leaders facilitate open and transparent communication channels, thus enabling employees to share their ideas, thoughts, and insights. This conducive environment fosters effective collaboration and problem-solving, ultimately enhancing task performance.

Further, digital leadership plays a crucial role in cultivating a culture of innovation and learning within an organization, positively influencing task performance [59]. Digital leaders prioritize the development of digital competence alongside an organization’s cultural values, equipping employees with the necessary knowledge, skills, and resources to explore and experiment with digital technologies. This integration empowers employees to creatively leverage digital tools and platforms, thus fostering innovative work and problem-solving approaches.

Additionally, digital leaders who effectively communicate and promote the digital vision inspire and motivate employees to embrace digitalization and think creatively about their roles and responsibilities [13]. By articulating a clear vision and demonstrating how digital technologies can enhance individual and team performance, leaders establish a sense of purpose and direction among employees, thus empowering them to excel in their tasks.

From a contextual performance perspective, digital leaders’ ability to increase employees’ willingness to exhibit selfless behavior, along with their transformational visions, fosters team spirit and facilitates goal attainment, ultimately enhancing contextual performance [59,60,61]. The technological competencies and effective utilization of information resources of digital leaders will positively influence task and contextual performance [62,63,64]. Consequently, digital leadership is expected to be a significant determinant of employee performance by leveraging digital technologies, thus fostering a culture of innovation and promoting the digital vision. Through enhanced communication, collaboration, knowledge-sharing, and creativity, digital leaders contribute to improved task performance. Moreover, their ability to inspire and motivate employees and instill a sense of purpose further enhances contextual performance. Accordingly,

**Hypothesis** **H2.***Digital leadership is positively and significantly related to employee performance: (a) task performance and (b) contextual performance*.

### 2.4. Effects of Individual Creativity on Employee Performance

Creativity, which entails the generation of novel and valuable ideas, is widely acknowledged as a critical driver of employee performance, particularly in roles and industries where innovation plays a pivotal role in attaining a competitive advantage [65,66]. Empirical research consistently demonstrates a positive association between individual creativity and employee performance, encompassing task and contextual performance.

For example, Oldham and Cummings [67] conducted a study involving research and development (R&D) professionals and found that individual creativity significantly correlated with high levels of employee performance. Similarly, Zhou and Shalley [68] identified that employees exhibiting elevated levels of creativity were more inclined to manifest superior performance. These findings underscore the positive impact of individual creativity on employee performance.

The mechanisms underlying the relationship between individual creativity and employee performance can be explicated through various factors. First, creative individuals possess a proclivity for generating innovative solutions to problems, contributing to enhanced performance [69]. Their ability to think divergently and offer fresh perspectives enables creative individuals to approach tasks and challenges with unique insights, ultimately fostering improved performance outcomes.

Second, individuals with high levels of creativity tend to be more engaged and motivated in their work, consequently enhancing their performance [70]. The intrinsic motivation and passion associated with the creative process drive individuals to invest greater effort and dedication in their tasks, thus yielding higher employee performance. Hence,

**Hypothesis** **H3.**
*Individual creativity is positively and significantly related to employee performance: (a) task performance and (b) contextual performance.*


### 2.5. Mediating Role of Individual Creativity

Based on the belief that employee creativity is positively reflected in work outcomes (e.g., employee performance), there should be a significant focus on the determinants of employee creativity (e.g., digital leadership). For example, digital leaders look for new ways of using technologies and implementing new ideas flexibly and creatively [71]. By doing so, they awaken their subordinates’ creative potential and enhance their performance [72]. The current research was based on the assumption that digital leadership improves employee creativity, ultimately promoting employee performance. Our research model (see Figure 1) argues that the effect of digital leaders on employees’ employee performance can be direct and indirect (e.g., via employee creativity).

Leaders with digital skills (i) usually run their business through digital platforms, communicating digitally [20]; (ii) incorporate culture and digital competence [21,22]; and (iii) transfer the digital vision to internal and external shareholders [13], thus emerging as innovation pioneers and role models for their followers. Digital leaders use up-to-date real-world tools to unleash creativity and passion for learning [53]. These can enhance followers’ willingness to face and go beyond obstacles and search for new resources to accomplish work goals [73], thus promoting employee creativity [74] and increasing employee performance [72]. In addition, when employees become creative, they work more intelligently as they begin creating novel ways to cope with daily work problems that ultimately positively influence their overall employee performance [75]. Hence,

**Hypothesis** **H4.***Individual creativity mediates the relationship between digital leadership skills and employee performance: (a) task performance and (b) contextual performance*. 

### 2.6. Exploring the Impacts of Generational Differences among Managers 

The concept of “generation” has gained significant importance since its introduction by Karl Mannheim over 50 years ago. This interdisciplinary concept refers to “a series of birthdays of a group of people” and is founded on the idea that individuals are influenced by shared experiences, such as historical events, socioeconomic conditions, and cultural phenomena during adolescence and early adulthood [76,77,78,79]. These common experiences among individuals of a particular age at a specific point in time create similarities, such as attitudes, political orientations, and general dispositions, among those in the cohort [80]. In the era of digital transformation, understanding these generational differences is particularly important for digital leadership. Different generations may have different levels of digital literacy, comfort with digital technologies, and attitudes toward digital transformation. For instance, while younger generations may be more comfortable with using digital technologies and more open to digital transformation, older generations may have more experience and wisdom to guide the digital transformation process [81,82]. Therefore, effective digital leadership requires an understanding of these generational differences and the ability to leverage the strengths of each generation. Digital leaders need to create an inclusive digital culture that respects and values generational diversity, fosters intergenerational learning and collaboration, and leverages the unique strengths and talents of each generation to drive digital transformation and enhance organizational performance [45,57].

In the literature, various classifications exist to differentiate one generation from another. The widely accepted Lyons and Kuron [83] classification, for example, identifies four generations that are currently active in the professional world: Baby Boomers, Gen X, Gen Y, and Gen Z [84,85]. Baby Boomers, born just after World War II, represent the largest generation among the classified generations and are known for their workaholic tendencies and large egos [80,86]. Gen X, the children of Baby Boomers, is a transitional generation between older generations that value tradition and younger ones that readily embrace new and digital technologies. They prioritize long-term employment and personal lives over work and are less hierarchical and more individualistic than Baby Boomers [80]. Gen Y, a highly educated generation, grew up with greater access to technology and is more likely to take “digital” breaks from work. They are considered the “Net” generation and prioritize collaborative work. Managers should design new jobs, particularly more digitalized ones, to work more with others on teams, special projects, task forces, or committees to develop their interpersonal skills and reward them for effective performance in collaborative efforts [87]. Finally, Gen Z, also known as digital natives, is characterized by their reliance on mobile devices and visual media and has a unique relationship with technology that distinguishes them from previous generations [88].

Research indicates that the generational cohorts of managers are reflected in subordinate-level outcomes, e.g., references [89,90]. As generations become younger, their interest in and proficiency with digital technologies increase, and it is more likely for managers belonging to younger generations to be perceived as superior in terms of digital leadership. Consequently, we anticipate that the relationships between employees’ perceptions of their managers’ digital leadership, individual creativity, and employee performance will differ, depending on the generational cohorts of the managers, with these relationships being stronger for managers in younger generations.

## 3. Research Design 

### 3.1. Measures 

The present study employed multi-item scales borrowed from previous research to measure the constructs of interest. Respondents rated their agreement with each item on a 5-point Likert scale, where 1 denoted “strongly disagree” and 5 denoted “strongly agree.” To assess the generational differences among participants’ supervisors, we used a categorical scale that asked respondents to indicate the age range of their supervisors. Specifically, the scale was composed of three categories: Baby Boomers (coded as 1), Gen X (coded as 2), and Gen Y (coded as 3).

Based on an extant literature review and a focus group interview conducted on five middle-level managers employed in organizations that engaged in a digital transformation process during the COVID-19 pandemic, we consider digital leadership skills a three-dimensional construct composed of technology-based vision, digital communication skills, and competency. In order to measure digital leadership abilities, we used the visionary leadership scale developed by Polney [91], digital communication skills scale developed Sokolov et al. [92], and competency scale developed by Polney [91]. Technology-based vision scale and technology competence scale are composed of five items, while the digital communication skills scale involves seven items, e.g., “The vision for my department incorporates technology” (technology-based vision), “My manager use technology as a tool for communicating, interacting, and engaging with employees and customers” (digital communication skills), and “My manager is familiar with our current technology infrastructure in terms of networks, bandwidth, Wi–Fi, hardware, and software” (competency). 

We used an individual creativity scale of five items, adapted from Houghton and DiLiello [93], to measure the creative performance of individuals in the workplace, e.g., “I have the freedom to decide how my job tasks get done” and “My creative abilities are used to my full potential at work.” Two different scales were used to measure task performance and contextual performance. For task performance, a nine-item scale adapted from Kirkman and Rosen [94] was used. Examples of the items in the scale include “I make sure that products meet or exceed quality standards.” On the other hand, contextual performance was measured using a seven-item scale adapted from Jawahar and Carr [95]. Examples of items in the scale include “I perform my duties with extra special care” and “I always meet or beat deadlines for completing work.”

### 3.2. Sampling 

The aim of this paper is to investigate the interrelationships among digital leadership capabilities, individual creativity, and employee performance, considering generational differences among managers. A target sample of 1000 post-graduates who are actively working in Istanbul and have been employed at the same organization for at least three years were selected from the records of the Alumni Association of İstanbul Beykent University. Initially, the purpose of this study was explained to all 1000 potential participants via telephone, of which 550 agreed to take part. After receiving responses from the participants, 352 employees completed the survey; then, after careful examination, all incomplete returns with missing data were discarded, leaving 348 responses for analysis. The data-collection process spanned a duration of 10 months, commencing in March 2021 and concluding in February 2022. Ethical clearance for data collection was secured from the Istanbul Beykent University Social Sciences Ethics Committee through a decision rendered on 4 February 2021, and designated as number 112910. This study’s empirical foundation rests upon data obtained from a convenience sample comprising participants with educational qualifications at the university diploma level or higher.

Among the participants, 192 were male (55%), and 198 (57%) were married. The mean age of the participants was 32 years (±1.73). In terms of work arrangements, 73 participants were engaged in remote work (21%), 161 were classified as hybrid workers (46%), and 114 were working on-site at the company (33%). Regarding industry distribution, 147 participants (42%) belonged to the manufacturing sector, 182 (52%) were affiliated with the service industry, and 19 (5%) were associated with the information technologies (IT) industry. This study also revealed that 55% of the participants’ supervisors belonged to Gen X, 29% belonged to Gen Y, and 16% belonged to Baby Boomers. 

G*Power 3.1.9.7 software was used to determine the sample size required for the analyses. The sample size analysis assumed a desired power of 80% [96]. The actual power determined was 0.95; the total sample size was 138 (α error probability was 0.005, 1-β error probability was 0.95, and the number of predictors was 5). The 348 samples in our research are considered sufficient.

### 3.3. Analysis

Our model was tested using the PLS-SEM technique in general and the PLS-MGA for the moderation hypothesis in particular for several reasons. First, PLS-SEM is preferred over maximum likelihood techniques, according to Fornell and Larcker [97], since it avoids many of the restrictive assumptions underlying them and ensures against improper solutions and factor indeterminacy. PLS-SEM does not require distributional assumptions regarding the indicators or error terms [98]; it also explicitly recognizes measurement errors. Second, PLS-SEM is insensitive to sample size considerations and is appropriate for any sample size over 30, unlike covariance-based SEM techniques [97,99]. Our sample is composed of 348 respondents (*n* = 348 employees); we also divide our sample into three subsamples for the MGA. Thus, we have smaller samples, which require PLS-SEM. Moreover, PLS handles reflective and formative constructs [99]. SEM provides several measures for evaluating a model’s goodness of fit. This allows researchers to assess the degree to which the model accurately represents the data. Hypothesis testing allows researchers to test multiple hypotheses simultaneously and to determine the strength of the relationships among variables in the model.

For the MGA, we employed the PLS-MGA technique, which allows researchers to test whether predefined data groups have significant differences in their group-specific parameter estimates (e.g., path coefficients), in this case, employees with Baby Boomer, Gen X, and Gen Y managers. Before performing the PLS-MGA testing, we first create test measurement models to ensure that the items forming the constructs used are currently valid and reliable. Then, PLS-MGA testing is operationalized via the SmartPLS 3 program [100].

### 3.4. Measurement Validation

In this study, we adopted reflective indicators for all our constructs, as suggested by Kleijnen, Ruyter, and Wetzels [101]. To assess the reliability and validity of the measurement instruments, we estimated a null model with no structural relationships and evaluated reliability using composite scale reliability (CR), Cronbach’s alpha, and average variance extracted (AVE). The results indicate that all measures exhibit satisfactory levels of reliability and convergent validity, with PLS-based CR and Cronbach alpha exceeding the threshold value of 0.70 and AVE exceeding the 0.50 threshold value for all first-order constructs. Further, standardized loadings of all measures on their respective constructs exceed 0.60, thus supporting their convergent validity. 

Table 1 showcases the relationships among the seven variables, reinforcing the notion of discriminant validity. The findings reveal that the constructs possess less mutual variance with other constructs in comparison with their corresponding constructs since each construct’s AVE surpasses the squared correlations among them. No construct intercorrelations surpass the square root of their respective AVEs, further validating their discriminant nature. Additionally, Table 1 includes a presentation of descriptive statistics.

Moreover, we examined the variance inflation factor (VIF) to assess the multicollinearity problem in the data. Aiken et al. [102] suggested that the values of VIF must be <10; we found that VIF values range between 1.118 and 2.320 for our study, depicting no issue of multicollinearity in the data.

Further, as a second-order variable, DL was estimated through a secondary factor analysis yielding three latent constructs: digital communication skills, digital integration skills, and vision toward digital technologies. Digital communication skills had six indicators; digital integration skills and social and vision toward digital technologies had five. Figure 2 shows the standardized regression loadings of those given three constructs.

### 3.5. Evaluating the Structural Model and Testing the Study Hypotheses

To estimate the indirect and main effects as well as perform multigroup analysis (MGA) and test the proposed model’s predictive power and hypotheses, we utilized the partial least-squares (PLS) approach and bootstrapping resampling method. We employed the SmartPLS 3.0 software program and computed T-statistics for all coefficients to determine their stability across subsamples and identify statistically significant links. The direction and impact of each hypothesized relationship were indicated by the path coefficients and their associated *t*-values. The proposed model is illustrated in Figure 1.

Table 2 shows the results of hypotheses, including paths, betas, and significance levels. Regarding the direct effects of DL, the results demonstrated that DL was significantly and positively associated with IC (β = 0.50; *p* < 0.01), thus supporting H1. However, the findings provide no empirical evidence in support of a direct relationship between DL and any performance outputs; hence, H2 is not supported. Moreover, the results showed that IC was significantly and positively associated with both TP (β = 0.80; *p* < 0.01) and CP (β = 0.78; *p* < 0.01), thus fully supporting H3.

Moreover, we also performed mediation analysis (see Table 3) to assess the mediating role of IC on the relationship between DL and employee performance (i.e., TP and CP). With the inclusion of mediating variable IC, the impact of DL on TP (β = −0.01; *p* > 0.05) and CP (β = 0.06; *p* > 0.05) becomes insignificant. However, the indirect effect of DL on TP (β = 0.40; *p* < 0.01) and on CP (β = 0.39; *p* < 0.01) was significant. These findings indicate that the relationships between DL and employee performance (i.e., TP and CP) are fully mediated by IC, thus supporting H4. The resulting model is presented in Figure 3.

### 3.6. Structural Model 

In order to validate the PLS-SEM approach, various quality scores, such as the coefficient of determination (R^2^) [103], the Q predictive validity (Q^2^), and NFI and SRMR [104], are considered. The R^2^ values of the endogenous constructs are used to evaluate the model fit and indicate how well data points fit a line or curve [103,104]. As suggested by [103], the categorization of R^2^ values is small (0.02 ≤ R^2^ < 0.13), medium (0.13 ≤ R^2^ < 0.26), or large (0.26 ≤ R^2^). The R^2^ statistic values of the endogenous constructs were used to assess model fit [104,105]. According to Table 4, individual creativity (R^2^ = 0.26), task performance (R^2^ = 0.64), and contextual performance (R^2^ = 0.66) all together had large effect sizes. The Q predictive validity of all our endogenous constructs was also considered sufficient. This finding means that the predictors in the models are able to explain the variance in the dependent variable.

Through the model fit criteria for PLS-SEM, we can observe two categories: the standardized root mean squared residual (SRMR) and the normed fit index (NFI). It is recommended that the SRMR value be equal to or less than 0.08 [106], while the NFI value must be greater than 0.90 [107]. Table 4 shows that, for our model, the SRME is 0.070, meeting the criteria. However, since we used a second-order composite variable, i.e., digital leadership, NFI is not estimated for our model. Accordingly, we conclude that the developed structural model has a satisfactory predictive power.

### 3.7. Moderation Effects for the Relationship between Groups (Generations) in the Structural Model

Confirmation of the moderating role of generations was achieved through multigroup analysis–partial least squares (MGA-PLS) test, which was supported by the Smart-PLS v.3 software program. MGA is a statistical technique used for comparing two or more groups to determine whether there are statistically significant differences in the estimate of parameters per group [108,109]. It enables researchers to assess differences in structural paths between multiple groups and is, therefore, an effective method for evaluating moderating effects through various relationships and structural paths [110]. The results of the MGA-PLS test are presented in Table 5, where three groups were tested: employees with Baby Boomer managers, employees with Gen X managers, and employees with Gen Y managers.

Interestingly, the results appeared close to tracks in all three groups. First, the **β** values for the DL to IC, DL to TP, and DL to CP paths of the first group are, respectively, 0.481 (*p* < 0.01), 0.090 (*p* > 0.05), and 0.013 (*p* > 0.05). On the other hand, for the employees with Gen X managers, **β** values for the DL to IC, DL to TP, and DL to CP paths are, respectively, 0.499 (*p* < 0.01), −0.040 (*p* > 0.05), and 0.030 (*p* > 0.05), while paths for the employees with the Gen Y managers are 0.501 (*p* < 0.01), 0.036 (*p* > 0.05), and 0.142 (*p* < 0.05). As a result, based on PLS-MGA, there is only a statistically significant difference regarding the direct effects of DL; further, the **β** value of the DL to CP path is significantly lower for employees with Baby Boomer managers than those with Gen Y managers (−0.129, *p* < 0.05). 

Second, regarding the association of IC on employee performance, the **β** values for the IC to TP and IC to CP paths of the first group are, respectively, 0.681 (*p* < 0.01) and 0.791 (*p* < 0.01). Meanwhile, for employees with Gen X managers, **β** values for IC to TP and IC to CP paths are, respectively, 0.790 (*p* < 0.01) and 0.778 (*p* < 0.01); moreover, those paths for the employees with employees with Gen Y managers are 0.876 (*p* < 0.01) and 0.779 (*p* < 0.01). Thus, based on PLS-MGA, again, there is only a statistically significant difference concerning the reflections of IC on performance outputs; moreover, **β** value of the IC to TP path is significantly lower for employees with Baby Boomer managers than those with Gen Y managers (−0.195, *p* < 0.05).

Finally, regarding the indirect effects of DL on performance outputs through IC, the **β** values for the DL to TP and DL to CP paths of the first group are, respectively, 0.327 (*p* < 0.01) and 0.380 (*p* < 0.01). Meanwhile, for employees with Gen X managers, **β** values for DL to TP and DL to CP paths are, respectively, 0.395 (*p* < 0.01) and 0.389 (*p* < 0.01), while those paths for employees with Gen Y managers are 0.438 (*p* < 0.01) and 0.390 (*p* < 0.01). Accordingly, PLS-MGA results do not provide empirical support for the statistical differences among the three groups for the indirect relationships.

## 4. Discussion 

This study investigated digital leadership from a generational perspective and demonstrated its impact on employee-level outcomes through empirical evidence. The study results show that digital leadership significantly influences employee creativity and, in turn, impacts employee performance. This study offers four key contributions to the literature.

First, this study reveals that the perception of digital leadership among employees toward their managers has a significant and positive relationship with individual creativity. The findings emphasize the importance of effective digital leadership in promoting creativity and innovation in the workplace. Effective digital leadership provides employees with the necessary resources and tools to be innovative and fosters a culture that values creativity and experimentation [111]. Additionally, employees may feel more comfortable sharing their creative ideas with managers who demonstrate effective digital leadership, as they are more likely to be receptive to new ideas and willing to use technology to facilitate innovation [112]. 

Second, this study shows that individual creativity has a positive relationship with task and contextual performance. Creative employees are more likely to engage in behaviors that contribute to a positive work environment and enhance contextual performance, such as helping coworkers and participating in organizational activities [113]. Moreover, individual creativity is believed to improve task performance by enhancing problem-solving and idea generation, leading to better outcomes and higher-quality work [114].

Third, this research demonstrates that the relationship between digital leadership and employee performance is fully mediated by individual creativity. Employees who perceive their managers as effective digital leaders may be more likely to engage in creative thinking, leading to improved employee performance [115]. Hence, individual creativity plays a critical role in the relationship between digital leadership and employee performance.

Finally, this study indicates that the interrelationships between digital leadership perceptions of employees toward their managers, individual creativity, and employee performance almost do not differ due to the generations of those managers. The perceptions of employees toward their managers are crucial in this context, rather than the generations of those managers. While there may be some minor differences in how different generations of employees perceive their managers and respond to digital leadership approaches, these differences are not significant enough to undermine the overall relationship between digital leadership, creativity, and employee performance. This interesting finding raises doubts about the assumption that younger generations are the most proficient and innovative digital technology users. Consistent with the study by Fortunati et al. [116], the differences between generations concerning digital skills may not be as definite.

### 4.1. Managerial Contributions

This paper highlights the importance of managers developing strong digital leadership skills to promote creativity and innovation among employees. By creating a supportive and technology-savvy work environment, managers can encourage their employees to explore new ideas and approaches, leading to improved outcomes for an organization. Moreover, by fostering creative thinking and providing opportunities for idea generation, managers can enhance task and contextual performance, which can further contribute to organizational success. To achieve this, managers should take the necessary steps to foster a culture of creativity and provide the resources required to support creative thinking. 

Further, it is crucial for managers to adopt a digital leadership approach that emphasizes collaboration, communication, and innovation. This approach should create a supportive work environment that values employee ideas and encourages creativity. By doing so, managers can help all employees, regardless of their generation, to realize their full potential and contribute to an organization’s success. Overall, this paper provides valuable insights into the critical role that digital leadership plays in promoting creativity and innovation; it also highlights the importance of developing a supportive work environment that values and fosters creative thinking.

### 4.2. Limitations and Future Research

This paper discusses several limitations of this study and provides recommendations for future research. One major limitation is the use of cross-sectional data, which raises concerns about the direction and purpose of the relationship between the constructs [117]. To address this issue, it is suggested that future studies collect longitudinal data.

Another limitation of this study is the use of self-reported data, although this method is practically useful in many research contexts [118]. However, recent studies indicate that self-reports may not be as limiting as previously believed and may provide more precise estimates than behavioral measures [119].

Moreover, it is essential to acknowledge that the present study’s sample is limited to employees in Istanbul, Turkey. Istanbul holds a prominent position in Turkey’s economy and industry, which suggests that managers in Istanbul may possess greater proficiency in various domains, including those related to digital technologies, compared with their counterparts in other regions of the country. To enhance the generalizability of the findings, future studies should extend their samples to include employees from diverse regions.

We also recommend that future research incorporates firm-level outputs, such as firm innovativeness, competitiveness, and overall firm performance, into the research model. This expansion would provide a more comprehensive understanding of the impact of digital leadership on organizational success. By examining the relationship between digital leadership and these key organizational outcomes, researchers can elucidate how digital leadership influences firms’ overall performance and competitiveness.

Further, there is a need for additional research and exploration to advance the understanding of digital leadership in the context of the Revolution 5.0 era. This line of inquiry can contribute to identifying best practices, frameworks, and strategies that enhance the effectiveness of digital leadership and guide organizations in their digital transformation journey. As the business landscape becomes increasingly digitalized, leaders who continuously adapt and evolve their digital leadership capabilities can position their organizations at the forefront of the digital revolution. This, in turn, can drive sustainable growth and success in the dynamic and ever-evolving digital business environment.

In conclusion, this study acknowledges several limitations and suggests various recommendations for future research, including the collection of longitudinal data, the inclusion of other regions in the sample, and the addition of firm innovativeness, competitiveness, and firm performance as outputs. Despite those limitations, this study sheds light on a critical yet underexplored concept, i.e., digital leadership. The results indicate a positive association between digital leadership and individual creativity and suggest that individual creativity fully mediates the relationship between digital leadership and employee performance. Additionally, this study finds that the interrelationships among digital leadership, individual creativity, and employee performance remain consistent across generational cohorts of managers, suggesting that managers, regardless of their generation, are developing their digital skills.

## Figures and Tables

**Figure 1 behavsci-14-00003-f001:**
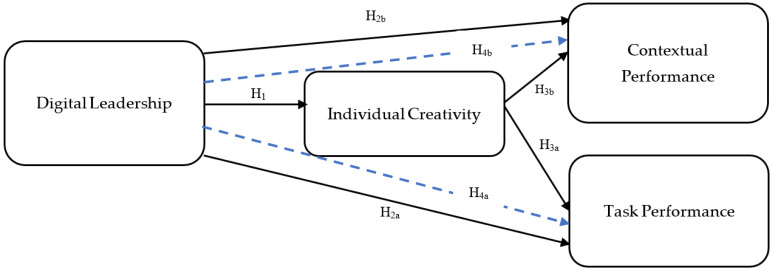
Research model.

**Figure 2 behavsci-14-00003-f002:**
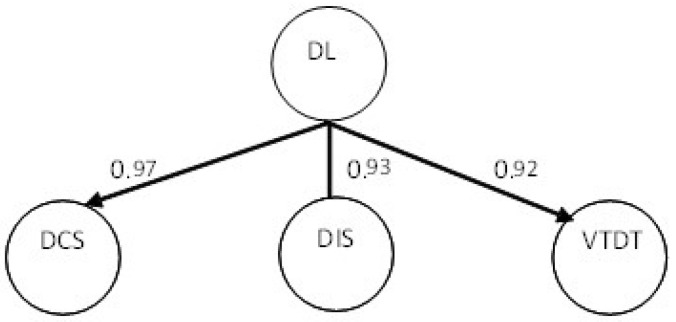
Second-order factor analysis of digital leadership. Note: DL = digital leadership; DCS = digital communication skills; DIS = digital integration skills; VTDT: vision toward digital technologies.

**Figure 3 behavsci-14-00003-f003:**
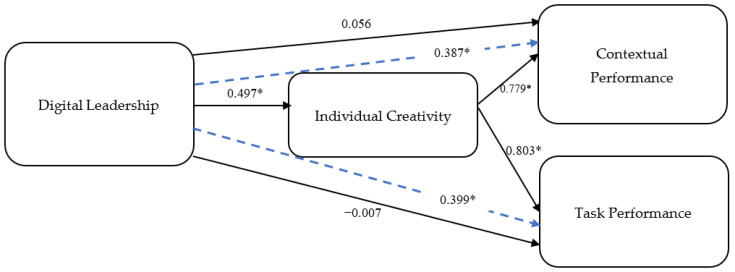
Resulting model. (* = *p* < 0.05).

**Table 1 behavsci-14-00003-t001:** Correlations and CR and AVE values.

Variables	1	2	3	4	5	6
CP	0.800					
IC	0.775	0.803				
DCS	0.366	0.427	0.834			
DIS	0.399	0.509	0.777	0.891		
TP	0.803	0.771	0.364	0.371	0.822	
VTDT	0.416	0.482	0.840	0.764	0.383	0.848
CR	0.926	0.879	0.931	0.951	0.936	0.927
AVE	0.641	0.645	0.695	0.795	0.676	0.719
α	0.906	0.817	0.910	0.935	0.920	0.902

Note: CP = contextual performance; IC = individual creativity; DCS = digital communication skills; DIS = digital integration skills; TP: task performance; VTDT: vision toward digital technologies.

**Table 2 behavsci-14-00003-t002:** Results of hypotheses.

Relationships			Path Coefficient (β)	Subhypotheses	Subresults	Hypotheses	Results
DL	→	IC	0.497 **	H1	Supported	H1	Supported
DL	→	TP	−0.007	H2a	Not Supported	H2	Not Supported
DL	→	CP	0.056	H2b	Not Supported		
IC	→	TP	0.803 **	H3a	Supported	H3	Supported
IC	→	CP	0.779 **	H3b	Supported		

Note: CP = contextual performance; IC = individual creativity; TP = task performance; DL = digital leadership. **** = *p* < 0.01.

**Table 3 behavsci-14-00003-t003:** Results of the mediating analyses.

Total Effect		Direct Effect		Indirect Effect			
Relationship	Path Coefficient (β)	Relationship	Path Coefficient (β)	Relationship	Path Coefficient (β)	BI [2.5%; 97.5%]	
DL→TP	0.392 **	DL→TP	−0.007	DL→IC→TP	0.399 **	0.304	0.492
DL→CP	0.443 **	DL→CP	0.056	DL→IC→CP	0.387 **	0.297	0.476

Note: CP = contextual performance; TP = task performance; IC = individual creativity; DL = digital leadership. *** p* < 0.01.

**Table 4 behavsci-14-00003-t004:** Structural model.

Endogenous Constructs	R^2^	Q^2^	SRMR
IC	0.247	0.154	0.071
TP	0.638	0.420	
CP	0.653	0.386	

Note: CP = contextual performance; TP = task performance; IC = individual creativity.

**Table 5 behavsci-14-00003-t005:** MGA results.

			Baby Boomers	Gen Xs	Gen Ys	BB vs. Gen X	BB vs. Gen Y	Gen X vs. Gen Y
Relationships			β	β	β	β	β	β
DL	→	IC	0.481 **	0.499 **	0.501 **	−0.019	−0.020	−0.001
DL	→	TP	0.090	−0.040	0.036	0.130	0.053	−0.076
DL	→	CP	0.013	0.030	0.142 *	−0.016	−0.129 *	−0.112
IC	→	TP	0.681 **	0.790 **	0.876 **	−0.109	−0.195 *	−0.086
IC	→	CP	0.791 **	0.778 **	0.779 **	0.013	0.012	−0.001
DL→IC→TP			0.327 **	0.395 **	0.438 **	−0.067	−0.111	−0.044
DL→IC→CP			0.380 **	0.389 **	0.390 **	−0.008	−0.010	−0.002

Note: CP = contextual performance; IC = individual creativity; TP = task performance; DL = digital leadership. ** p* < 0.05; *** p* < 0.01.

## Data Availability

The data presented in this study are available on request from the corresponding author.

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
