# Peer review of "Digital Leadership’s Influence on Individual Creativity and Employee Performance: A View through the Generational Lens"

_behavsci, 2023, doi:10.3390/bs14010003_

Round 1
Reviewer 1 Report (New Reviewer)
Comments and Suggestions for Authors
Hello,
The article presents a study on the relationship between digital leadership and performance, as well as the mediating role of creativity. The authors also add a multi-group analysis in order to detect the generational effect in a sample of 348 employees. The study is important especially for organizations which have seen a transformation of their work organization after COVID-19, and have favored a combination of face-to-face work with teleworking, and thus introduced remote management of their employees. The research topic is presented in a clear manner, and the literature review is quite in-depth presenting the most recent and salient results from previous studies. The research framework is justified as well as the four hypotheses. However, we noted some weaknesses related to the analyses, of which here are the main ones:
1. The relationship between creativity and performance presents regression betas that are too high (0.7 and 0.8), which indicates a multicollinearity problem
2. Add a statistical power analysis to be sure not to commit type II error
Good luck to the authors in their research work
Comments on the Quality of English LanguageN/A
Author Response
Dear Sir,
This letter outlines the revisions we have made to the manuscript published in your journal. We would like to share with you the corrections and additions we implemented based on the feedback received after the initial review.
- VIF Values: VIF values have been added to demonstrate that our analyses are not prone to multicollinearity issues.
- Power Analysis: A power analysis has been conducted.
We kindly request that our paper be reconsidered, and the revisions be reviewed. We appreciate your attention and contributions throughout this process.
Sincerely
Reviewer 2 Report (New Reviewer)
Comments and Suggestions for Authors
I believe this paper has been examined and verified by other reviewers that I do not find any critical issues in the article. However in abstract the authors mention multi group analysis (MGA) while in body text it refers as mediated group analysis (MGA). Be consistent, and check in the whole article, which one is correct?
Author Response
Dear Sir,
This letter outlines the revisions we have made to the manuscript published in your journal. We would like to share with you the corrections and additions we implemented based on the feedback received after the initial review.
- Multigroup Analysis (MGA) Clarification: Confusion regarding Multigroup Analysis (MGA) in the text has been rectified.
We kindly request that our paper be reconsidered, and the revisions be reviewed. We appreciate your attention and contributions throughout this process.
Sincerely
Reviewer 3 Report (New Reviewer)
Comments and Suggestions for Authors
In this paper, the authors attempted to examine the influence of Digital Leadership on individual creativity and employee performance. I believe this is an interesting topic in alignment with the rapid development of new technologies and the evolving organization of work. Digital leadership is indeed a leadership approach that highlights the use of digital technologies, data, and innovation to drive organizational success and transformation in the digital age. Therefore, digital leaders need to possess a combination of traditional leadership skills and a profound understanding of digital tools and strategies.
I think the paper addresses a current topic in a clear and effective manner with the appropriate tools. However, I recommend that the authors consider the following minor aspects before publication:
- I didn't understand why there are highlighted parts in yellow in the text.
- From line 68 to line 72 (page 2), it's not clear what the authors intend to convey.
- The authors have considered a sample of post-graduate workers with 348 final respondents. Given the observed period, it would be useful to clarify whether these workers work remotely or on-site at the company. This is also in line with the article's emphasis on the role of remote working in the introduction.
- To better clarify methods and results, I think it would be useful to indicate the type of companies in which these workers are employed (e.g., economic sector, manufacturing, services...) and the number of companies involved.
Author Response
Dear Sir,
This letter outlines the revisions we have made to the manuscript published in your journal. We would like to share with you the corrections and additions we implemented based on the feedback received after the initial review.
- Highlighted Areas: As part of this revision, improvements in the text have been highlighted in yellow to emphasize the recommendations from the previous reviews.
- Lines 68-72: The section from line 68 to line 72 on page 2 has been clarified.
- Participant Details: Details such as whether participants work remotely or on-site and the industry in which they are employed have been added. However, as we gathered the data individually, we do not have any information regarding the number of companies involved.
We kindly request that our paper be reconsidered, and the revisions be reviewed. We appreciate your attention and contributions throughout this process.
Sincerely
Round 2
Reviewer 1 Report (New Reviewer)
Comments and Suggestions for Authors
Hi,
The authors had responded to all our comments and suggestions indicated in the first report.
This manuscript is a resubmission of an earlier submission. The following is a list of the peer review reports and author responses from that submission.
Round 1
Reviewer 1 Report
Comments and Suggestions for Authors
Dear Authors,
Well done research paper. However, you mentioned that "In this study, we adopted reflective indicators for all our constructs, as suggested by Kleijnen, Ruyter, and Wetzels (2007)."
Kleijnen, M., De Ruyter, K., & Wetzels, M. (2007). An assessment of value creation in mobile service delivery and the moderating role of time consciousness. Journal of retailing, 83(1), 33-46.
When I related to their paper, I could not find the constructs/variables/items. They stated "This study focuses on the perceived utilitarian value of a new service delivery mode, the mobile channel. The authors develop a framework that incorporates three mode-specific benefits – time convenience, user control, and service compatibility – as well as two costs – perceived risk and cognitive effort – as antecedents of perceived value. "
Please check.
Reviewer 2 Report
Comments and Suggestions for Authors
Dear Author(s) of the manuscript proposal entitled "Digital Leadership's Influence on Individual Creativity and Employee Performance: A View Through the Generational Lens" sent to Behavioral Sciences Journal, please find below my concerns and recommendations.
You should know that before writing this review report, I conducted a self-documentation process. During this process I found that your manuscript proposal is similar (about 50%) to a material submitted to Beykent Universitesi. Please clarify, together with the editor of the journal, this issue. Is that material yours? Is it in a repository?
Regarding the content itself, please address the followings:
1. The Introduction chapter is in general well-written and it presents the research question between the rows 97 - 100. However, it doesn't clearly define and describe the research gap and the research goal. Please add a new paragraph where, based on the previous literature, you present these two important aspects.
2. After hyothesis 3, between the rows 255 - 265 you have the "Figure 1. Research Model". In my version of the pdf file, the figure seems to be "wrapped" and the arrows between the rows 263 - 264 points to... nowhere. I think it is an editing issue. Please revise and correct it. Also, I recommend you to put the hypotheses symbols in the arrows, so that the readers understand the proposed model.
3. The chapter "2. Theoretical background and hypothesis development" should be improved by including the following relevant resources: https://doi.org/10.1108/CG-10-2019-0306, https://doi.org/10.3390/electronics12132857, https://doi.org/10.3390/su12145771, https://doi.org/10.3390/systems10050134. These articles will be a strong support for your research context.
4. In the subchapter "3.2. Sampling" you present the sample used in this study. Please also present the following data: the period of time the survey was conducted, and if the respondents were awarded for their participation. If yes, how?
5. The Table 1 is entitled "Correlations, CR and AVE Values." Thus, I understand that in the first part of the table, you present the correlations values. As I see, on each row is one of the 6 variables (CP, IC, DCS, DIS, TP, VTDT) and on the columns the same variables are presented (1, 2, 3, 4, 5, 6). I don't understand how is it possible to have correlations between the same variables (CP and 1, IC and 2, ...) which are different from 1. Please revise this aspect.
6. In "Table 3. Results of the mediating analyses" there are some values with green color. Is there any signification for this choice?
7. At the end of the chapter 3, I recommend you to include a visual figure (similar to figure 1) with the values draw on the arrows.
Dear Author(s),
Please consider all the above remarks as being constructive recommendations in order to improve the general quality of your manuscript proposal.
Kind Regards!